# Remimazolam's clinical application and safety: A signal detection analysis based on FAERS data and literature support

Gang Ye[1], Luqin Ding[2], Qingbo Zhou[2]*

1 Department of Pain Medicine, Shaoxing People's Hospital, Shaoxing, China, 2 Department of Internal Medicine, Shaoxing Yuecheng People's Hospital, Shaoxing, China

* zhouqingbo@zcmu.edu.cn

## Abstract

This study aimed to evaluate the adverse event profile of remimazolam, a novel ultra-short-acting benzodiazepine, with a focus on its safety in the respiratory, cardiovascular, and immune systems across diverse patient populations. We analyzed adverse event reports from the FAERS database over a defined period, performing signal detection using the proportional reporting ratio (PRR) and the reporting odds ratio (ROR), and contextualized the findings with a concurrent literature review. Remimazolam demonstrated a strong signal for hypoventilation. In the cardiovascular system, it was associated with serious adverse events, including cardiac and cardiorespiratory arrest, particularly in high-risk patients. Furthermore, we detected significant signals for severe hypersensitivity reactions, such as anaphylactic shock and laryngeal edema, while signals in other systems were less pronounced but remained clinically significant. Given that the study population was predominantly elderly, and considering the serious nature of the identified signals, its potential for adverse events necessitates vigilant monitoring. Future research should focus on clarifying risks within specific high-risk groups to establish optimized safety protocols.

## Introduction

Remimazolam, a novel ultra-short-acting benzodiazepine, is distinguished by its rapid onset and swift recovery profile [1]. Compared to traditional sedatives like propofol and midazolam, remimazolam offers greater hemodynamic stability and a lower risk of respiratory depression. These properties make it a valuable agent for procedural sedation, general anesthesia, and other settings requiring rapid patient recovery [2]. Moreover, its distinct metabolic pathway is thought to lower the incidence of adverse reactions, particularly in elderly patients and those with comorbidities [3].

With the expanding clinical use of remimazolam, especially in high-risk populations, a systematic evaluation of its safety profile has become crucial. While

**Data availability statement:** All relevant data underlying the results presented in this study are publicly available from the FDA Adverse Event Reporting System (FAERS) database. The data used in this study cover the period from [specific time range, e.g., Q2 2020 to Q1 2023] and can be accessed via the FDA website at https://www.fda.gov/drugs/questions-and-answers-fdas-adverse-event-reporting-system-faers.

**Funding:** The author(s) received no specific funding for this work.

**Competing interests:** The authors have declared that no competing interests exist.

pre-market clinical trials provide foundational safety data, their limited sample sizes and short follow-up periods may not detect rare or delayed adverse events in a broad population [4,5]. Therefore, post-marketing surveillance using real-world data is essential for a comprehensive safety assessment. Indeed, pharmacovigilance studies using real-world databases are instrumental for assessing the post-marketing safety profiles of various therapeutics. These studies complement findings from pre-approval clinical trials, as seen in studies on sildenafil [6] and everolimus [7].

This study analyzes data from the FDA Adverse Event Reporting System (FAERS) to investigate potential safety signals associated with remimazolam. FAERS is a global pharmacovigilance database maintained by the U.S. FDA, which contains millions of spontaneous adverse event reports from patients, healthcare professionals, and pharmaceutical companies [8,9]. The utility of FAERS for such pharmacovigilance, and its capacity for signal detection in diverse therapeutic areas, is well-documented in studies concerning topotecan [10], acetylsalicylic acid [11], anastrozole [12], and vinca alkaloids [13].

The extensive coverage of FAERS allows for the identification of infrequent or delayed adverse event signals that might be missed in clinical trials [14]. By systematically analyzing remimazolam-related reports in this database, our study aims to provide a more comprehensive evidence base for its real-world safety. Ultimately, this work seeks to inform safer prescribing strategies, especially for high-risk patient groups.

## Methodology

### Data source and processing

This study is based on data from the U.S. Food and Drug Administration's (FDA) Adverse Event Reporting System (FAERS), a public database of spontaneous adverse event reports submitted by patients, healthcare professionals, and pharmaceutical manufacturers. For our analysis, we queried the entire available database, spanning from the first quarter of 2004 to the fourth quarter of 2023, to extract all reports associated with remimazolam. The resulting dataset, which covered reports from the second quarter of 2020 to the first quarter of 2023, formed the basis of our study [9].

Data processing was performed in several stages. First, we removed duplicate reports by retaining only the most recent entry for each unique case ID. Next, data from different tables were linked using the primaryid field. We then standardized drug names using the MedEx-UIMA system and cleaned the dataset by correcting or removing records with abnormal age or weight entries. Finally, we isolated all reports where remimazolam was listed as a suspect drug and extracted key variables, including patient demographics and report details. The complete workflow for data selection and processing is illustrated in Fig 1.

### Signal detection analysis

To identify potential safety signals, we employed a multi-algorithm approach to ensure robust and comprehensive detection. Four distinct signal detection algorithms were used: the Reporting Odds Ratio (ROR), the Proportional Reporting Ratio (PRR),

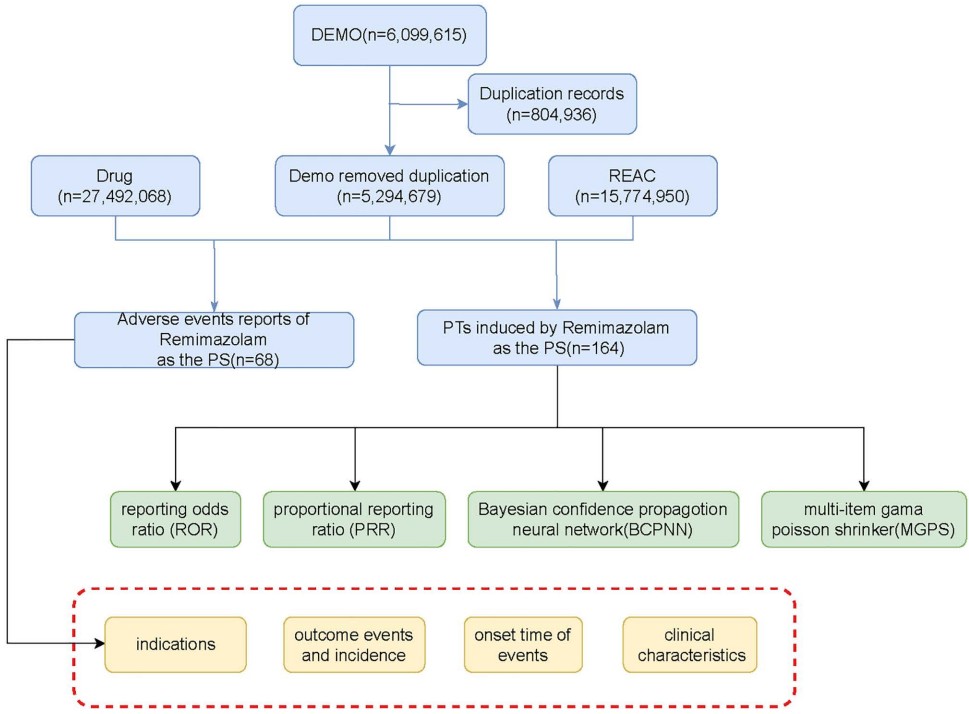

**Fig 1. Flow diagram of the selection process for remimazolam-related adverse events.** The diagram illustrates the stepwise filtering of reports from the FAERS database, showing the number of adverse event (AE) records identified and excluded at each stage to arrive at the final analytical dataset.

the Bayesian Confidence Propagation Neural Network (BCPNN), and the Multi-Item Gamma Poisson Shrinker (MGPS). This combination allowed for the detection of both common and rare adverse event signals. The specific formulas and criteria for these algorithms are detailed in Fig 2.

The criteria for signal detection for each algorithm were as follows:

ROR: A signal was defined as the lower limit of the 95% confidence interval (CI) being greater than 1 [15].

PRR: A signal required a PRR value $\geq 2$ and a corresponding chi-square value $\geq 4$ [16].

BCPNN: A signal was present if the lower limit of the 95% CI of the Information Component (IC) was greater than 0 [17].

MGPS: This empirical Bayesian method was used to detect signals for rare events [18].

All calculations were performed using R software (version 4.1.3). The aggregated results from these algorithms formed the basis for our evaluation of remimazolam's adverse event profile.

## Signal screening and classification

This threshold (N $\geq$ 3) is a commonly adopted criterion in pharmacovigilance to enhance the stability of disproportionality measures and to reduce the likelihood of spurious signals arising from isolated case reports [19].

All identified signals were coded and classified using the Medical Dictionary for Regulatory Activities (MedDRA). We focused our analysis on Preferred Terms (PT) within each System Organ Class (SOC). This systematic classification allowed us to identify the primary organ systems affected and to provide a clear framework for analyzing remimazolam's safety risks [15].

(i) **ROR Algorithm**

$$ROR = \frac{ad}{bc}$$

$$95\%CI = e^{\ln(ROR) \pm 1.96\sqrt{\frac{1}{a}+\frac{1}{b}+\frac{1}{c}+\frac{1}{d}}}$$

**Signal detection criterion:** The lower bound of the 95% CI > 1, and $N \geq 3$.

(ii) **PRR Algorithm**

$$PRR = \frac{a(c+d)}{c(a+b)}$$

$$\chi^2 = \frac{(a+b+c+d)(ad-bc)^2}{(a+b)(c+d)(a+c)(b+d)}$$

**Signal detection criterion:** $PRR \geq 2$, $\chi^2 \geq 4$, and $N \geq 3$.

(iii) **BPCNN Algorithm**

$$IC = \log_2 \frac{a(a+b+c+d)}{(a+c)(a+b)}$$

$$95\%CI = E(IC) \pm 2 \times \sqrt{V(IC)}$$

**Signal detection criterion:** $IC025 > 0$ (IC025 is the lower bound of the 95% CI).

(iv) **EBGM Algorithm**

$$EBGM = \frac{a(a+b+c+d)}{(a+c)(a+b)}$$

$$95\%CI = e^{\ln(EBGM) \pm 1.96\sqrt{\frac{1}{a}+\frac{1}{b}+\frac{1}{c}+\frac{1}{d}}}$$

**Signal detection criterion:** $EBGM05 > 2$ (EBGM05 is the lower bound of the 95% CI).

**Fig 2. Signal detection algorithms and their corresponding criteria.** This figure outlines the formulas and specific thresholds used for the four signal detection algorithms (ROR, PRR, BCPNN, and MGPS) employed in this study.

## Results

### Overview of dataset characteristics

A total of 68 adverse event reports associated with remimazolam were analyzed. The number of reports increased annually between 2020 and 2023, peaking in 2022, which accounted for 35.3% of the total reports (Table 1). Reports for male patients (55.9%) were more frequent than for female patients (27.9%), with gender unspecified in 16.2% of cases. The median patient age was 67.5 years and the median weight was 65.5 kg, indicating a predominantly elderly study population. Physicians submitted the majority of reports (79.4%), followed by pharmacists (19.1%). The primary route of administration was intravenous (88.2%). Regarding outcomes, most events were classified as "other serious" (74.7%), while 12.7% were "life-threatening" and 2.8% were fatal. The median time-to-onset was 0 days, suggesting that most adverse events occurred shortly after drug administration.

### SOC-level analysis

Table 2 presents the signal detection results at the System Organ Class (SOC) level. A signal was considered significant if it met the predefined criteria: a Reporting Odds Ratio (ROR) with a 95% CI lower bound > 1, or a Proportional Reporting Ratio (PRR) ≥ 2 with a chi-square (χ²) value ≥ 4. All analyzed signals were based on at least three case reports.

**Table 1. Demographic and report characteristics of the study population.**

| Variable | Category/Value | Total n (%) |
|---|---|---|
| **Year** | | |
| 2020 | 6 (8.82) | |
| 2021 | 23 (33.82) | |
| 2022 | 24 (35.29) | |
| 2023 | 15 (22.06) | |
| **Sex** | | |
| Female | 19 (27.94) | |
| Male | 38 (55.88) | |
| Unknown | 11 (16.18) | |
| **Age (years)** | Median (IQR) | 67.50 (54.50, 75.75) |
| **Weight (kg)** | Median (IQR) | 65.50 (56.50, 69.68) |
| **Reporter** | | |
| Physician | 54 (79.41) | |
| Pharmacist | 13 (19.12) | |
| Consumer | 1 (1.47) | |
| **Reported countries** | | |
| Other | 68 (100.00) | |
| **Route of Administration** | | |
| Intravenous | 60 (88.24) | |
| Other | 8 (11.76) | |
| **Outcomes** | | |
| Other serious | 53 (74.65) | |
| Life-threatening | 9 (12.68) | |
| Hospitalization | 7 (9.86) | |
| Death | 2 (2.82) | |

Abbreviations: N, Number of reports; IQR, Interquartile Range.

Significant disproportional reporting was identified for several SOCs. The strongest signal was for Immune System Disorders (17 cases), with an ROR of 10.02 (95% CI 6.07, 16.56) and a PRR of 9.09. This suggests a reporting frequency for these events with remimazolam that is approximately 10 times higher than the background rate. This SOC includes critical Preferred Terms (PTs) like anaphylactic shock, which generated an even more pronounced signal at the PT level (Table 3).

Cardiac Disorders (22 cases) also generated a significant signal (ROR 7.56; 95% CI 4.83, 11.85), with a reporting frequency approximately 7.6 times higher than expected. Respiratory, Thoracic and Mediastinal Disorders (27 cases) presented a signal with an ROR of 4.08 (95% CI 2.70, 6.16), indicating a reporting frequency about 4 times higher than expected. This SOC contains crucial events like hypoventilation.

Other SOCs that met the signal criteria included Vascular Disorders (13 cases; ROR 4.41) and Investigations (28 cases; ROR 3.16). These findings highlight specific system organ classes where remimazolam is associated with a disproportionately high frequency of adverse event reports, underscoring the need for targeted clinical monitoring.

## Analysis results of specific adverse reactions

At the Preferred Term (PT) level, a detailed analysis revealed strong safety signals for specific adverse events, with full details presented in Table 3. This granular analysis identified critical risks, some of which were not apparent at the broader SOC level.

**Table 2. Signal detection results at the System Organ Class (SOC) level.**

| SOC | Case Reports | ROR (95% CI) | PRR (95% CI) | chisq | IC (IC025) | EBGM (EBGM05) |
|---|---|---|---|---|---|---|
| Immune system disorders | 17 | 10.02(6.07, 16.56) | 9.09(5.79, 14.27) | 123.75 | 3.18(2.48) | 9.09(5.97) |
| Cardiac disorders | 22 | 7.56(4.83, 11.85) | 6.68(4.51, 9.89) | 108.45 | 2.74(2.11) | 6.68(4.59) |
| Vascular disorders | 13 | 4.41(2.5, 7.78) | 4.14(2.44, 7.03) | 31.6 | 2.05(1.26) | 4.14(2.58) |
| Respiratory, thoracic and mediastinal disorders | 27 | 4.08(2.7, 6.16) | 3.57(2.51, 5.08) | 52.41 | 1.84(1.26) | 3.57(2.53) |
| Investigations | 28 | 3.16(2.11, 4.75) | 2.79(2, 3.89) | 34.32 | 1.48(0.91) | 2.79(1.99) |
| Injury, poisoning and procedural complications | 26 | 1.27(0.84, 1.93) | 1.23(0.86, 1.75) | 1.26 | 0.3(−0.29) | 1.23(0.86) |
| Skin and subcutaneous tissue disorders | 7 | 0.81(0.38, 1.73) | 0.82(0.4, 1.69) | 0.29 | −0.28(−1.31) | 0.82(0.44) |
| Nervous system disorders | 5 | 0.39(0.16, 0.95) | 0.41(0.17, 0.97) | 4.6 | −1.29(−2.46) | 0.41(0.19) |
| General disorders and administration site conditions | 11 | 0.32(0.17, 0.59) | 0.36(0.2, 0.64) | 15 | −1.46(−2.3) | 0.36(0.22) |
| Gastrointestinal disorders | 4 | 0.29(0.11, 0.78) | 0.31(0.12, 0.81) | 6.88 | −1.71(−3) | 0.31(0.13) |

Abbreviations: SOC, System Organ Class; Case Reports, Number of individual adverse event reports; ROR, Reporting Odds Ratio; CI, Confidence Interval; PRR, Proportional Reporting Ratio; chisq, Chi-square value; IC, Information Component; IC025, Lower limit of the 95% confidence interval for the IC; EBGM, Empirical Bayes Geometric Mean; EBGM05, Lower limit of the 90% confidence interval for the EBGM.

**Investigations.** Within this SOC, Blood Pressure Decreased (8 cases) and Oxygen Saturation Decreased (6 cases) generated substantial signals. The signal for Blood Pressure Decreased was particularly strong (ROR 51.98), indicating a reporting frequency over 50 times higher than expected. Similarly, Oxygen Saturation Decreased was reported over 30 times more frequently than its background rate (ROR 33.33), highlighting remimazolam's potential to impact these key physiological parameters.

**Cardiac disorders.** Serious cardiac events produced prominent signals. Cardio-Respiratory Arrest (3 cases; ROR 42.92) and Cardiac Arrest (4 cases; ROR 25.62) were reported over 40 and 25 times more frequently than expected, respectively. These findings underscore a significant safety concern regarding remimazolam's impact on cardiac function.

**Respiratory, thoracic, and mediastinal disorders.** Hypoventilation (3 cases) produced the most powerful safety signal for an adverse drug reaction in this study, with an ROR of 450.81. This indicates a reporting frequency over 400 times higher than the background rate, identifying a critical safety risk. Additionally, Bronchospasm (5 cases) generated a very strong signal (ROR 189.02). These results necessitate close respiratory monitoring during remimazolam administration.

**Immune system disorders.** Severe hypersensitivity reactions were frequently reported. Anaphylactic Shock (6 cases) was reported over 100 times more frequently than expected (ROR 105.97), and Anaphylactic Reaction (9 cases) was also associated with a strong signal (ROR 68.03).

**Injury, poisoning and procedural complications.** The most statistically prominent signal in the entire analysis was for Vascular Access Site Occlusion (11 cases), which had an exceptionally high ROR of 36,584.93. While this extraordinary disproportionality highlights a potential issue, its clinical interpretation requires further investigation to rule out confounding factors related to administration procedures or reporting artifacts.

## Discussion

### Overview of pharmacological properties and study findings on remimazolam

Remimazolam, a novel ultra-short-acting benzodiazepine, was developed to overcome the limitations of traditional sedatives like propofol and midazolam, namely their associated risks of respiratory depression and hemodynamic instability. The rapid metabolism and short half-life of remimazolam are intended to mitigate these risks and preserve hemodynamic stability. However, our analysis of the FAERS database indicates that remimazolam can still elicit significant adverse events affecting the respiratory and cardiovascular systems. Our findings offer new insights into remimazolam's real-world

**Table 3. Signal detection results for specific adverse events at the Preferred Term (PT) level.**

| SOC | PT | Case Reports | ROR (95% CI) | PRR (95% CI) | chisq | IC (IC025) | EBGM (EBGM05) |
|---|---|---|---|---|---|---|---|
| Investigations | blood pressure decreased | 8 | 51.98(25.54, 105.8) | 49.49(25.42, 96.37) | 380.28 | 5.63(4.66) | 49.47(27.29) |
| Investigations | oxygen saturation decreased | 6 | 33.33(14.75, 75.32) | 32.15(14.68, 70.42) | 181.21 | 5.01(3.91) | 32.14(16.24) |
| Investigations | heart rate increased | 3 | 12.07(3.85, 37.83) | 11.87(3.88, 36.28) | 29.9 | 3.57(2.14) | 11.87(4.56) |
| Cardiac disorders | cardio-respiratory arrest | 3 | 42.92(13.7, 134.53) | 42.16(13.79, 128.85) | 120.54 | 5.4(3.97) | 42.14(16.2) |
| Cardiac disorders | cardiac arrest | 4 | 25.62(9.5, 69.09) | 25.01(9.57, 65.34) | 92.28 | 4.64(3.36) | 25.01(10.9) |
| Cardiac disorders | tachycardia | 3 | 14.3(4.56, 44.81) | 14.06(4.6, 42.97) | 36.43 | 3.81(2.38) | 14.06(5.41) |
| Vascular disorders | hypotension | 8 | 16.66(8.19, 33.91) | 15.9(8.17, 30.96) | 112 | 3.99(3.02) | 15.89(8.77) |
| Vascular disorders | flushing | 3 | 16.1(5.14, 50.46) | 15.83(5.18, 48.38) | 41.71 | 3.98(2.55) | 15.82(6.08) |
| Respiratory, thoracic and mediastinal disorders | hypoventilation | 3 | 450.81(143.5, 1416.19) | 442.58(144.81, 1352.65) | 1315.75 | 8.78(7.35) | 440.56(169.06) |
| Respiratory, thoracic and mediastinal disorders | bronchospasm | 5 | 189.02(77.54, 460.76) | 183.28(77.37, 434.16) | 904.88 | 7.52(6.34) | 182.94(86.8) |
| Immune system disorders | anaphylactic shock | 6 | 105.97(46.88, 239.55) | 102.13(46.63, 223.69) | 600.41 | 6.67(5.58) | 102.02(51.56) |
| Immune system disorders | anaphylactic reaction | 9 | 68.03(34.73, 133.25) | 64.35(34.37, 120.49) | 561.43 | 6.01(5.09) | 64.31(36.64) |
| Skin and subcutaneous tissue disorders | erythema | 4 | 8.74(3.24, 23.57) | 8.55(3.27, 22.34) | 26.74 | 3.1(1.81) | 8.55(3.73) |
| Injury, poisoning and procedural complications | vascular access site occlusion | 11 | 36584.93(18061.01, 74107.53) | 34131.13(17528.18, 66460.64) | 277096.74 | 14.62(13.67) | 25192.28(13955.91) |
| General disorders and administration site conditions | drug interaction | 4 | 11.19(4.15, 30.18) | 10.94(4.19, 28.58) | 36.2 | 3.45(2.17) | 10.94(4.77) |

Abbreviations: SOC, System Organ Class; PT, Preferred Term; Case Reports, Number of individual adverse event reports; ROR, Reporting Odds Ratio; CI, Confidence Interval; PRR, Proportional Reporting Ratio; chisq, Chi-square value; IC, Information Component; IC025, Lower limit of the 95% confidence interval for the IC; EBGM, Empirical Bayes Geometric Mean; EBGM05, Lower limit of the 90% confidence interval for the EBGM.

safety profile. The respiratory and cardiovascular signals we identified are particularly noteworthy. Existing literature suggests that such risks can be more pronounced in high-risk populations, such as elderly patients and individuals with pre-existing comorbidities [3], which underscores the need for heightened caution when administering the drug to these vulnerable patient groups. These findings offer new insights into remimazolam's real-world safety profile and underscore the need for heightened caution when administering the drug to vulnerable patient groups.

## Discussion of multi-system adverse reactions of remimazolam

This pharmacovigilance study systematically evaluated remimazolam's real-world safety profile using data from the FAERS database. Our analysis identified significant adverse event signals across several key organ systems, most notably the respiratory, cardiovascular, and immune systems [20].

## Respiratory adverse reactions

A primary safety concern with any sedative is the potential for respiratory depression. Although remimazolam was designed to have a more favorable respiratory safety profile, our analysis revealed a powerful signal for Hypoventilation.

This finding demonstrates that the risk of respiratory depression with remimazolam persists. The clinical implications of this are significant. For context, this risk may be heightened in high-risk groups, such as elderly patients and individuals with pre-existing respiratory disease, as suggested by other clinical studies [21].

This finding has significant clinical implications. Our findings underscore the critical need for vigilant respiratory monitoring during remimazolam administration, particularly in vulnerable populations or during prolonged sedation. Clinicians must be prepared to provide supplemental oxygen or mechanical ventilation if necessary [22,23]. Therefore, future research should aim to establish optimal dosing regimens that maximize sedative efficacy while minimizing respiratory adverse events [24].

### Cardiovascular adverse reactions

Our analysis also identified significant safety signals for cardiovascular adverse events. We found a disproportionately high reporting frequency for Cardiac Arrest and Cardio-Respiratory Arrest. While our study could not stratify risk by patient history, these findings are particularly concerning because existing literature suggests that patients with pre-existing conditions like heart failure may be more vulnerable to such events [25].

While remimazolam is often selected for its favorable hemodynamic profile, these findings align with clinical reports that it can still induce hypotension and bradycardia, particularly in vulnerable patients [26]. Therefore, in high-risk settings such as cardiac surgery, a cautious approach is essential. This includes vigilant hemodynamic monitoring, preparedness to manage acute cardiac events, and consideration of a "start low, go slow" dosing strategy to mitigate the risk of serious cardiovascular complications [27].

### Immune system adverse reactions

Our analysis revealed strong signals for severe hypersensitivity reactions, most notably Anaphylactic Shock and Anaphylactic Reaction. This finding is consistent with existing case reports which indicate that remimazolam can elicit life-threatening immune responses, including laryngeal edema [28,29].

These results carry significant clinical implications. A thorough evaluation of a patient's allergy history, particularly regarding sensitivities to other benzodiazepines, is essential prior to administration. Furthermore, clinicians must ensure vigilant monitoring during drug infusion and have emergency medications and equipment, such as epinephrine, immediately available to manage any potential acute allergic event.

### Comparative safety profile and other clinically relevant events

While our analysis did not generate strong signals for all adverse events, some still warrant clinical consideration. For instance, events like hypotension, nausea, and postoperative delirium, though not meeting the threshold for a strong signal in our data, are known risks that may be more pronounced in elderly patients or those on concomitant medications [30]. Postoperative delirium, in particular, remains a significant concern in the elderly due to its potential to delay recovery [31].

When contextualized against other sedatives, remimazolam's safety profile is distinct. Its primary advantage is superior hemodynamic stability, with a lower incidence of hypotension compared to agents like propofol, making it valuable in high-risk surgeries [32]. However, this must be balanced against the specific clinical scenario. For prolonged deep sedation, propofol may offer more predictable control [33]. Conversely, compared to midazolam, remimazolam's rapid clearance provides a key benefit by facilitating faster recovery and potentially reducing the risk of postoperative delirium, a notable advantage in elderly populations [34].

Ultimately, while remimazolam presents a favorable alternative to traditional sedatives, these findings emphasize that no single agent is universally optimal. The choice of sedative necessitates a careful, individualized risk-benefit assessment, considering both patient-specific factors and procedural demands to ensure safety and efficacy [35].

## Study limitations

This study has several inherent limitations. First, its reliance on the FAERS database, a spontaneous reporting system, means it is subject to underreporting and various reporting biases [36]. Second, while disproportionality analysis can identify statistical signals, it cannot establish causality or calculate the true incidence of adverse events [37]. Finally, the relatively small sample size of remimazolam reports in our dataset (N = 68) may limit the statistical power to detect very rare events, a common challenge in post-marketing surveillance for newer drugs [38]. Therefore, the safety signals identified here should be seen as important hypotheses requiring validation in future controlled studies.

## Conclusion

By systematically analyzing real-world data from the FAERS database, this study provides a comprehensive post-marketing safety evaluation of remimazolam, filling a critical gap in the existing literature, which has primarily focused on pre-market clinical trials.

Our findings confirm remimazolam's known benefits but also highlight several significant safety signals that warrant clinical attention. Most notably, we identified powerful signals for respiratory depression (e.g., Hypoventilation), serious cardiac events (e.g., Cardiac Arrest), and severe hypersensitivity reactions (e.g., Anaphylactic Shock). These findings underscore the need for vigilant monitoring and risk mitigation strategies during remimazolam use. Given that such risks may be amplified in vulnerable populations, as suggested by the broader clinical literature, our results highlight the importance of careful patient selection and monitoring, especially when treating elderly patients or those with significant comorbidities.

The signals identified in this pharmacovigilance study provide a critical foundation for future investigation. To advance from signal detection to confirmed risk assessment, it is essential that these findings are validated through large-scale, multicenter prospective studies. Such research is necessary to quantify adverse event incidence more precisely and establish definitive clinical guidelines. Furthermore, long-term follow-up studies utilizing real-world data from electronic health records (EHRs) will be crucial for assessing remimazolam's long-term safety profile. Ultimately, this study provides a vital evidence base to inform safer clinical practice and guide future research, ensuring that the therapeutic benefits of remimazolam can be maximized while its risks are carefully managed.

## Supporting information

**S1 File.  R script for FAERS data mining and analysis.** This file contains the complete R script used to perform the pharmacovigilance analysis. The script uses the faersR package to filter the FAERS database for adverse event reports associated with Remimazolam and to calculate key disproportionality metrics (e.g., ROR, PRR). The dplyr and openxlsx packages are subsequently used for data manipulation and to export the final result tables.
(R)

## Author contributions

**Conceptualization:** Gang Ye.

**Data curation:** Qingbo Zhou.

**Methodology:** Luqin Ding.

**Project administration:** Gang Ye.

**Supervision:** Gang Ye.

**Writing – original draft:** Qingbo Zhou.

**Writing – review & editing:** Qingbo Zhou.

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
