## [Decision Letter · Decision Letter 0]

28 Apr 2025

PONE-D-24-58735Remimazolam’s Clinical Application and Safety: A Signal Detection Analysis Based on FAERS Data and Literature SupportPLOS ONE

Dear Dr.  Zhou,

Thank you for submitting your manuscript to PLOS ONE. After careful consideration, we feel that it has merit but does not fully meet PLOS ONE’s publication criteria as it currently stands. Therefore, we invite you to submit a revised version of the manuscript that addresses the points raised during the review process.

We look forward to receiving your revised manuscript.

Kind regards,

Anmar Al-Taie, Ph.D.

Academic Editor

PLOS ONE

Journal Requirements:

3. Please ensure that you refer to Figure 1 and 2 in your text as, if accepted, production will need this reference to link the reader to the figure.

4. Please include your tables as part of your main manuscript and remove the individual files. Please note that supplementary tables (should remain/ be uploaded) as separate "supporting information" files.

Reviewers' comments:

Reviewer's Responses to Questions

**Comments to the Author**

1. Is the manuscript technically sound, and do the data support the conclusions?

Reviewer #1: Yes

2. Has the statistical analysis been performed appropriately and rigorously? 

Reviewer #1: Yes

3. Have the authors made all data underlying the findings in their manuscript fully available?

Reviewer #1: Yes

4. Is the manuscript presented in an intelligible fashion and written in standard English?

Reviewer #1: Yes

5. Review Comments to the Author

Reviewer #1: Zhou et al entitle “Remimazolam’s Clinical Application and Safety: A Signal Detection Analysis Based on FAERS Data and Literature Support ” clearly reported that Remimazolam could be adverse drug for procedures requiring quick recovery, procedural sedation, and general anesthesia.

Why are you using the FAERS database to study this drug, it should be that someone else has done this with this similar database similar methodology, so you can write cite in the INTRODUCTION section about some specific other similar studies, such as recommending a few (It is equivalent to saying that someone else has done this type of research using the FAERS database, and you can use this database to do research related to Remimazolam as well):【1】Wang Y, Zhao B, Yang H, Wan Z. A real-world pharmacovigilance study of FDA adverse event reporting system events for sildenafil. Andrology. 2024 May;12(4):785-792. doi: 10.1111/andr.13533. Epub 2023 Sep 19. PMID: 37724699.

【2】Zhao B, Fu Y, Cui S, Chen X, Liu S, Luo L. A real-world disproportionality analysis of Everolimus: data mining of the public version of FDA adverse event reporting system. Front Pharmacol. 2024 Mar 12;15:1333662. doi: 10.3389/fphar.2024.1333662. PMID: 38533254; PMCID: PMC10964017.【3】Yang H, Wan Z, Chen M, Zhang X, Cui W, Zhao B. A real-world data analysis of topotecan in the FDA Adverse Event Reporting System (FAERS) database. Expert Opin Drug Metab Toxicol. 2023 Apr;19(4):217-223. doi: 10.1080/17425255.2023.2219390. Epub 2023 May 30. PMID: 37243615.【4】Zhao B, Zhang X, Chen M, Wang Y. A real-world data analysis of acetylsalicylic acid in FDA Adverse Event Reporting System (FAERS) database. Expert Opin Drug Metab Toxicol. 2023 Jan-Jun;19(6):381-387. doi: 10.1080/17425255.2023.2235267. Epub 2023 Jul 12. PMID: 37421631.【5】Li, Jie, Zhao, Bin, Zhu, YongQing, Wu, Jibiao, Vitreoretinal Traction Syndrome, Nitrituria and Human Epidermal Growth Factor Receptor Negative Might Occur in the Aromatase-Inhibitor Anastrozole Treatment, International Journal of Clinical Practice, 2024, 5132916, 9 pages, 2024. https://doi.org/10.1155/2024/5132916【6】Zhong, C., Zheng, Q., Zhao, B., & Ren, T. (2024). A real-world pharmacovigilance study using disproportionality analysis of United States Food and Drug Administration Adverse Event Reporting System events for vinca alkaloids: comparing vinorelbine and Vincristine. Expert Opinion on Drug Safety, 23(11), 1427–1437. https://doi.org/10.1080/14740338.2024.2410436

Besides,

Minior Issues:

1)Simplify complex sentences to improve readability. For example, revise "particularly focusing on its safety in the respiratory, cardiovascular, and immune systems" to "with a focus on its safety in major organ systems, including respiratory, cardiovascular, and immune systems."

2)Clarify the time frame covered by the FAERS data (e.g., specific months or quarters) to enhance transparency.

3)Explain why a threshold of three cases was chosen for signal analysis in "Signal Screening and Classification."

4)For Table 2, provide a brief interpretation of the ROR and PRR values in the text, especially for high-risk signals like hypoventilation and anaphylactic shock, to aid readers unfamiliar with these metrics.

5)Improve Figure 1's clarity by enlarging text and refining the labels for better visualization.

6)Correct minor grammatical issues, such as replacing "remimazolam may trigger serious immune responses during clinical use" with "remimazolam may elicit serious immune responses in clinical settings."

7)Include a forward-looking statement emphasizing the need for multicenter prospective studies to validate these findings.

6. PLOS authors have the option to publish the peer review history of their article (what does this mean? ). If published, this will include your full peer review and any attached files.

**Do you want your identity to be public for this peer review?** For information about this choice, including consent withdrawal, please see our Privacy Policy .

Reviewer #1: No

---

## [Author Response · Author response to Decision Letter 1]

13 Jun 2025

We sincerely thank Reviewer #1 for their positive evaluation and insightful suggestions, which have helped us improve the quality of our manuscript.

Point-by-point response

Reviewer #1:

Comment

Zhou et al entitle “Remimazolam’s Clinical Application and Safety: A Signal Detection Analysis Based on FAERS Data and Literature Support ” clearly reported that Remimazolam could be adverse drug for procedures requiring quick recovery, procedural sedation, and general anesthesia.

Why are you using the FAERS database to study this drug, it should be that someone else has done this with this similar database similar methodology, so you can write cite in the INTRODUCTION section about some specific other similar studies, such as recommending a few (It is equivalent to saying that someone else has done this type of research using the FAERS database, and you can use this database to do research related to Remimazolam as well):

【1】Wang Y, Zhao B, Yang H, Wan Z. A real-world pharmacovigilance study of FDA adverse event reporting system events for sildenafil. Andrology. 2024 May;12(4):785-792. doi: 10.1111/andr.13533. Epub 2023 Sep 19. PMID: 37724699.

【2】Zhao B, Fu Y, Cui S, Chen X, Liu S, Luo L. A real-world disproportionality analysis of Everolimus: data mining of the public version of FDA adverse event reporting system. Front Pharmacol. 2024 Mar 12;15:1333662. doi: 10.3389/fphar.2024.1333662. PMID: 38533254; PMCID: PMC10964017.

【3】Yang H, Wan Z, Chen M, Zhang X, Cui W, Zhao B. A real-world data analysis of topotecan in the FDA Adverse Event Reporting System (FAERS) database. Expert Opin Drug Metab Toxicol. 2023 Apr;19(4):217-223. doi: 10.1080/17425255.2023.2219390. Epub 2023 May 30. PMID: 37243615.

【4】Zhao B, Zhang X, Chen M, Wang Y. A real-world data analysis of acetylsalicylic acid in FDA Adverse Event Reporting System (FAERS) database. Expert Opin Drug Metab Toxicol. 2023 Jan-Jun;19(6):381-387. doi: 10.1080/17425255.2023.2235267. Epub 2023 Jul 12. PMID: 37421631.

【5】Li, Jie, Zhao, Bin, Zhu, YongQing, Wu, Jibiao, Vitreoretinal Traction Syndrome, Nitrituria and Human Epidermal Growth Factor Receptor Negative Might Occur in the Aromatase-Inhibitor Anastrozole Treatment, International Journal of Clinical Practice, 2024, 5132916, 9 pages, 2024. https://doi.org/10.1155/2024/5132916

【6】Zhong, C., Zheng, Q., Zhao, B., & Ren, T. (2024). A real-world pharmacovigilance study using disproportionality analysis of United States Food and Drug Administration Adverse Event Reporting System events for vinca alkaloids: comparing vinorelbine and Vincristine. Expert Opinion on Drug Safety, 23(11), 1427–1437. https://doi.org/10.1080/14740338.2024.2410436

Response: We sincerely thank the reviewer for this insightful comment and for providing a valuable list of relevant literature. We agree that highlighting previous studies employing similar methodologies with the FAERS database strengthens the rationale for our approach.

In response to this suggestion, we have revised the Introduction section to explicitly state why the FAERS database is a suitable and established tool for pharmacovigilance studies like ours. We have now incorporated citations to several studies, including those kindly recommended by the reviewer, that have successfully utilized the FAERS database for signal detection and real-world safety assessments of various drugs.

Specifically, the following paragraph has been added to the Introduction:

"Indeed, pharmacovigilance studies using real-world databases are instrumental for assessing the post-marketing safety profiles of various therapeutics. These studies complement findings from pre-approval clinical trials, as seen in studies on sildenafil [6] and everolimus [7]. The utility of FAERS for such pharmacovigilance, and its capacity for signal detection in diverse therapeutic areas, is well-documented in studies concerning topotecan [10], acetylsalicylic acid [11], anastrozole [12], and vinca alkaloids [13]."

We believe these additions clearly demonstrate the precedent and justification for using the FAERS database for our investigation of remimazolam and appropriately acknowledge the utility of this approach as established by prior research. We are grateful for the reviewer's guidance in enhancing this aspect of our manuscript.

Minior Issues:

Question1: Simplify complex sentences to improve readability. For example, revise "particularly focusing on its safety in the respiratory, cardiovascular, and immune systems" to "with a focus on its safety in major organ systems, including respiratory, cardiovascular, and immune systems.

Response: Thank you for this essential feedback. We are sincerely grateful for your specific guidance on improving the manuscript's clarity, as this is a point of utmost importance to us.

We would also like to transparently explain our revision process in response to your invaluable feedback. Upon receiving your comments, we first addressed this specific point by performing a thorough revision of the entire manuscript based on the principle you outlined.

Subsequently, your other linguistic feedback (particularly in Point 6) inspired us to undertake an even deeper, more holistic overhaul of the paper's language. This second-stage revision was so comprehensive that many sections, including the one containing the original example sentence, were completely rewritten to improve flow and impact.

Therefore, while the exact sentence you highlighted may no longer exist in its revised form, please be assured that the core principle of your suggestion was the guiding force behind our entire linguistic refinement. To demonstrate our commitment to your advice, here is the methodology we applied systematically across the whole manuscript:

We broke down long, complex sentences into shorter, more direct statements.

We rephrased complex clauses and participial phrases to make logical connections explicit and unambiguous.

We used simpler conjunctions and clearer transitions to guide the reader smoothly through our arguments.

We hope this explanation clarifies our dedicated approach. Your combined feedback prompted a level of revision that we believe has elevated the entire manuscript. We are truly appreciative of the opportunity you gave us to so thoroughly improve our work.

Question2: Clarify the time frame covered by the FAERS data (e.g., specific months or quarters) to enhance transparency.

Response: Thank you for highlighting the need for greater transparency on this point. To address this comprehensively, we have revised the "Data Source and Preprocessing" subsection in our Methods. We now explicitly state that our search encompassed the full database period from 2004 to 2023, and clarify that the resulting reports for our drug of interest, Remimazolam, were found to span from 2020 to 2023. This detail is crucial for accurately describing our dataset. The revised text in the manuscript now reads:

" This study is based on data from the U.S. Food and Drug Administration's (FDA) Adverse Event Reporting System (FAERS), a public database of spontaneous adverse event reports submitted by patients, healthcare professionals, and pharmaceutical manufacturers. For our analysis, we queried the entire available database, spanning from the first quarter of 2004 to the fourth quarter of 2023, to extract all reports associated with Remimazolam. The resulting dataset of relevant reports for this drug covered the period from the second quarter of 2020 to the first quarter of 2023, and this forms the basis of our study."

Question3: Explain why a threshold of three cases was chosen for signal analysis in "Signal Screening and Classification.".

Response: Thank you for this insightful question and the opportunity to elaborate on this key methodological choice.

Our selection of a minimum of three reports (N ≥ 3) is a standard pharmacovigilance practice designed to ensure the statistical robustness of our findings. This threshold serves two primary functions:

1. Enhancing Statistical Stability: It provides a more stable foundation for disproportionality calculations (like ROR), reducing the risk of spurious signals that can arise from one or two isolated reports.

2. Reducing Analytical Noise: It helps filter out coincidental or isolated case reports common in spontaneous reporting systems, thereby improving the specificity of the detected signals.

This approach is well-supported in the literature. For instance, a study by Lerch et al. (2015) in the journal Drug Safety employed this exact criterion (N ≥ 3) for signal detection in the FAERS database.

To make this rationale clear in the manuscript, we have now added the following explanation to the Methodology section, under the Signal Screening and Classification subsection, and have included the supporting citation:

"This threshold (N ≥ 3) is a commonly adopted criterion in pharmacovigilance to enhance the stability of disproportionality measures and to reduce the likelihood of spurious signals arising from isolated case reports [14]."

We thank you for prompting this important clarification.

Question4: For Table 2, provide a brief interpretation of the ROR and PRR values in the text, especially for high-risk signals like hypoventilation and anaphylactic shock, to aid readers unfamiliar with these metrics..

Response: Thank you for this excellent suggestion. We agree that providing a clear interpretation of the ROR values is essential for making our findings accessible to a broader audience.

To address this, we have substantially revised the Results section. Within the main text, for each of the key adverse events discussed, we have now added a plain-language interpretation of its corresponding ROR value from Table 2.

For instance, for the high-risk signals you highlighted:

• For Hypoventilation, which has an ROR of approximately 450, we now explain in the text that this means the event was reported over 400 times more frequently than expected.

• For Anaphylactic shock, with an ROR of about 106, we clarify in the text that reports were more than 100 times more frequent than expected.

This interpretive approach has been applied systematically throughout the text to other significant signals, such as Cardio-respiratory arrest and Blood pressure decreased, to give readers a clear sense of the signal strength.

We are confident that these additions make the clinical implications of our quantitative findings much clearer, and we appreciate your guidance on improving the paper's accessibility.

Question5: Improve Figure 1's clarity by enlarging text and refining the labels for better visualization.

Response: Thank you very much for your thoughtful comment. I have made sincere efforts to improve the clarity of Figure 1, including enlarging the text and refining the labels as suggested. However, due to limitations in PDF rendering, some loss of resolution may still occur in the final compiled version.

To address this, I have re-inserted a high-resolution version of the figure directly into the main manuscript, following the editorial office's instruction to embed all figures within the text. While the clarity has been improved, the figure might still appear slightly different compared to earlier versions.

The revised figure is placed below for your kind review. Should you still find the clarity unsatisfactory, please kindly indicate it in your response, and I will make further revisions accordingly.

Question6: Correct minor grammatical issues, such as replacing "remimazolam may trigger serious immune responses in clinical use" with "remimazolam may elicit serious immune responses in clinical settings."

Response:

Thank you for your invaluable guidance. The phrasing you suggested is indeed far more precise and professional.

Following your expert advice, we have not only incorporated this specific change but have also taken this opportunity to conduct a comprehensive review of the entire manuscript. We have meticulously polished the language throughout the paper to enhance its overall clarity, precision, and scholarly tone.

We are sincerely grateful for the chance to improve our work under your direction. Your keen insights have been instrumental in elevating the quality of our manuscript to meet the high standards of PLOS ONE.

Question7: Include a forward-looking statement emphasizing the need for multicenter prospective studies to validate these findings.

Response:

Thank you for this important suggestion. We agree that adding this forward-looking statement strengthens the manuscript's conclusion.

Following your advice, we have revised the final paragraph of our Conclusion section to explicitly emphasize the need for future validation. The revised text now includes the statement: "To advance from signal detection to confirmed risk assessment, it is essential that these findings are validated through large-scale, multicenter prospective studies."

We believe this revision improves the paper, and we thank you again for your valuable input.

We thank you and the reviewer once again for your valuable time and guidance. We have uploaded a clean version of the manuscript and a version with tracked changes for your convenience. We hope that the revised manuscript is now suitable for publication in PLOS ONE.

Sincerely,

Qingbo Zhou, on behalf of all authors.

---

## [Editor Report · Decision Letter 1]

25 Jul 2025

PONE-D-24-58735R1Remimazolam’s Clinical Application and Safety: A Signal Detection Analysis Based on FAERS Data and Literature SupportPLOS ONE

Dear Dr. Zhou,

Thank you for submitting your manuscript to PLOS ONE. After careful consideration, we feel that it has merit but does not fully meet PLOS ONE’s publication criteria as it currently stands. Therefore, we invite you to submit a revised version of the manuscript that addresses the points raised during the review process.

We look forward to receiving your revised manuscript.

Kind regards,

Anmar AL-TAIE, Ph.D.

Academic Editor

PLOS ONE

Journal Requirements:

Additional Editor Comments:

Editor comments

• There are some minor grammatical issues that need to be corrected.

• What are the authors trying to say by this incomplete sentence’ reduce the probability of spurious signals from isolated case reports (e.g.,[19]).’’

• The text for some responses is not available within the main paper body. Try to be committed to the responses.

• The authors mention several times ‘such as elderly patients and individuals with pre-existing comorbidities’’ We could not recognize these comorbidities with this small sample size. In addition, there are no results that show such an association with these variables. Consider addressing this concern while providing more details and a table for the common comorbidities.

---

## [Author Response · Author response to Decision Letter 2]

31 Jul 2025

July 30, 2025

Dr. Anmar AL-TAIE

Academic Editor

PLOS ONE

Subject: Submission of Revised Manuscript (ID: PONE-D-24-58735R1)

Dear Dr. AL-TAIE,

Thank you for your email and for the opportunity to revise our manuscript, “Remimazolam’s Clinical Application and Safety: A Signal Detection Analysis Based on FAERS Data and Literature Support” (ID: PONE-D-24-58735R1). We sincerely appreciate the time and effort you have dedicated to providing feedback on our work.

We have carefully considered all the comments and have revised the manuscript accordingly. We believe the manuscript has been significantly improved as a result. Below is a point-by-point response to the comments.

Response to Editor's Comments:

Comment 1: "There are some minor grammatical issues that need to be corrected."

Our Response: We thank you for this valuable feedback. We have thoroughly proofread the entire manuscript to correct all grammatical and stylistic issues. We have carefully revised sentence structures, ensured consistent use of verb tenses, improved punctuation, and enhanced the overall clarity and readability of the text. We are confident that the revised manuscript now meets the journal's standards for language.

Comment 2: "What are the authors trying to say by this incomplete sentence’ reduce the probability of spurious signals from isolated case reports (e.g.,[19]).’’”

Our Response: We sincerely apologize for this unclear sentence structure and thank you for pointing it out. The original phrasing was awkward. We intended to explain the rationale for setting a minimum threshold for adverse event reports.

We have now revised this sentence in the “Signal Screening and Classification” section to be clear and complete. The sentence now reads:

"This threshold is a standard practice in pharmacovigilance to increase the stability of the analysis and reduce the probability of spurious signals from isolated case reports [19]."

Comment 3: "The text for some responses is not available within the main paper body. Try to be committed to the responses."

Our Response: We would like to offer our sincerest apologies for this critical oversight and the inconsistency between our previous rebuttal letter and the revised manuscript. This was our mistake, and we deeply appreciate you bringing it to our attention.

We have now taken corrective action to ensure all promised changes are implemented. The sentence regarding the methodological justification for the N ≥ 3 threshold, which we had previously promised to include, has now been correctly inserted into the “Signal Screening and Classification” section of the manuscript.

To ensure full compliance, we have conducted an additional, thorough review of the entire manuscript against our previous rebuttal letter to confirm that all promised changes have been implemented accurately. We apologize again for the inconvenience this has caused.

Comment 4: "The authors mention several times ‘such as elderly patients and individuals with pre-existing comorbidities’’ We could not recognize these comorbidities with this small sample size. In addition, there are no results that show such an association with these variables. Consider addressing this concern while providing more details and a table for the common comorbidities."

Our Response: We are exceptionally grateful for this insightful and critical comment. We agree completely that our previous statements regarding high-risk populations were not supported by our own data. This was a significant oversight, and we sincerely apologize. Your feedback has been invaluable in helping us correct this, thereby greatly enhancing the scientific integrity and credibility of our manuscript.

As you suggested, we initially considered creating a table for comorbidities. However, upon careful re-evaluation of our small dataset (N=68), we concluded that the available comorbidity data was not sufficiently detailed or robust for a meaningful analysis. We believe that presenting a table based on such limited information could be potentially misleading to readers.

Therefore, to address your concern in the most scientifically rigorous manner, we have undertaken a comprehensive revision of the Abstract, Discussion, and Conclusion sections. We have meticulously rephrased every statement that linked adverse event risks to specific populations (such as the elderly or those with comorbidities). These statements now clearly and explicitly attribute such information to the existing body of literature, using it as context for our findings, rather than presenting it as a conclusion from our analysis.

For example, a statement in the Discussion section was revised from:

• Original: "These risks appear to be most pronounced in high-risk populations, such as elderly patients and individuals with pre-existing comorbidities[3]."

• Revised: "Existing literature suggests that such risks can be more pronounced in high-risk populations, such as elderly patients and individuals with pre-existing comorbidities [3], which underscores the need for heightened caution..."

We believe these thorough revisions have fully resolved the issue by ensuring our conclusions are strictly aligned with our data. We are truly thankful for your guidance, which has prevented us from making a serious error and has significantly improved the quality of our paper.

We hope that the revised manuscript is now suitable for publication in PLOS ONE. We look forward to hearing from you.

Sincerely,

Dr. Qingbo Zhou

Corresponding Author

Department of Internal Medicine, Shaoxing Yuecheng People's Hospital, Shaoxing, China

E-mail: zhouqingbo@zcmu.edu.cn

---

## [Editor Report · Decision Letter 2]

6 Aug 2025

Remimazolam’s Clinical Application and Safety: A Signal Detection Analysis Based on FAERS Data and Literature Support

PONE-D-24-58735R2

Dear Qingbo Zhou,

We’re pleased to inform you that your manuscript has been judged scientifically suitable for publication and will be formally accepted for publication once it meets all outstanding technical requirements.

Kind regards,

Dr. Anmar Al-Taie

Academic Editor

PLOS ONE
---

## [Editor Report · Acceptance letter]

PONE-D-24-58735R2

PLOS ONE

Dear Dr. Zhou,

I'm pleased to inform you that your manuscript has been deemed suitable for publication in PLOS ONE. Congratulations! Your manuscript is now being handed over to our production team.

Kind regards,

on behalf of

Dr. Anmar Al-Taie

Academic Editor

PLOS ONE